# Investigation of the *EIL/EIN3* Transcription Factor Gene Family Members and Their Expression Levels in the Early Stage of Cotton Fiber Development

**DOI:** 10.3390/plants9010128

**Published:** 2020-01-20

**Authors:** Haron Salih, Shoupu He, Hongge Li, Zhen Peng, Xiongming Du

**Affiliations:** 1Institute of Cotton Research of Chinese Academy of Agricultural Sciences (ICR, CAAS), State Key Laboratory of Cotton Biology, Anyang 455000, Henan, China; salih234@yahoo.com (H.S.); heshoupu@caas.cn (S.H.); lihongge@caas.cn (H.L.); pengzhen0501@163.com (Z.P.); 2Department of Crop Science, College of Agriculture, Zalingei University, P.O. BOX 6, Central Darfur, Sudan; 3Zhengzhou Research Base, State Key Laboratory of Cotton Biology, Zhengzhou University, Zhengzhou 450001, China

**Keywords:** identification, gene structure, phylogeny, *EIL/EIN3*, *Cis*-element, developmental stages

## Abstract

The ethylene-insensitive3-like/ethylene-insensitive3 (*EIL/EIN3*) protein family can serve as a crucial factor for plant growth and development under diverse environmental conditions. *EIL/EIN3* protein is a form of a localized nuclear protein with DNA-binding activity that potentially contributes to the intricate network of primary and secondary metabolic pathways of plants. In light of recent research advances, next-generation sequencing (NGS) and novel bioinformatics tools have provided significant breakthroughs in the study of the *EIL/EIN3* protein family in cotton. In turn, this paved the way to identifying and characterizing the *EIL/EIN3* protein family. Hence, the high-throughput, rapid, and cost-effective meta sequence analyses have led to a remarkable understanding of protein families in addition to the discovery of novel genes, enzymes, metabolites, and other biomolecules of the higher plants. Therefore, this work highlights the recent advance in the genomic-sequencing analysis of higher plants, which has provided a plethora of function profiles of the *EIL/EIN3* protein family. The regulatory role and crosstalk of different metabolic pathways, which are apparently affected by these transcription factor proteins in one way or another, are also discussed. The ethylene hormone plays an important role in the regulation of reactive oxygen species in plants under various environmental stress circumstances. *EIL/EIN3* proteins are the key ethylene-signaling regulators and play important roles in promoting cotton fiber developmental stages. However, the function of *EIL/EIN3* during initiation and early elongation stages of cotton fiber development has not yet been fully understood. The results provided valuable information on cotton *EIL/EIN3* proteins, as well as a new vision into the evolutionary relationships of this gene family in cotton species.

## 1. Introduction

The gaseous ethylene phytohormone plays an important role in the regulation of different biological processes such as plant growth, plant development, and stress responses [1,2,3,4,5,6,7]. A large number of molecular studies have associated the ethylene pathway with signal transduction by triggering downstream ethylene response genes [8]. Furthermore, the *EIL/EIN3* genes also act as a foundation for ethylene connections with other signals, such as the crosstalk between ethylene and other hormones, light signaling, as well as various abiotic and biotic stress responses [9,10]. In cotton, two pathways of ethylene related genes were found to be associated with fiber yield [11]. Ethylene-insensitive3 pathways such as (*EIN3*) and *EIN3*-like (*EIL*) proteins are one of the main transcription factors [12,13,14]. The *EIL/EIN3* proteins were indicated to be involved in the expression of the GCC-box-binding domain of *Arabidopsis* ethylene-response factors [15]. The *EIN3/EIL* protein family is a small group of transcription factors in higher plants. Several transcription factor genes of *EIL/EIN3* proteins have been investigated in *Arabidopsis* [16,17], tobacco [18], tomato [19], cucumber [20,21,22], and rice [23]. Plant specific *EIL/EIN3* proteins have highly conserved amino acid sequences at the N-termini, and are located in the nuclei compartment [16]. Some important structural forms were shown in the first half of the *EIL/EIN3* protein sequences, such as highly acidic N-terminus amino acid regions, 5-basic amino acid groups, and proline-rich regions [17]. It is known that the *EIN3/EIL* protein family has multiple functions in plant growth and development. In *Arabidopsis*, *EIL3* acts as the central transcriptional regulator of plant sulfur response and metabolism process [24]. Overexpression of *EIL3* restores the sulfur limitation responseless morphologies of *Arabidopsis* mutants (slim1 mutants) [24]. In addition, it was found that *EIN3* significantly associates with the expression of chlorophyll biosynthesis genes [25]. Transcriptome analysis of *Arabidopsis* mutants found several *EIN3*-regulated genes that were coregulated by the other transcription factors such as PIFs (light signaling) and RHD6 (root hair development) [26], which indicates co-regulation of *EIL/EIN3*-activated transcription by certain developmental processes and environmental conditions. In tomato, overexpression of *EIN3*-binding F-box protein2 gene prompted elongated fruit shape and delayed fruit development and ripening [27]. In cotton, ethylene plays a vital role in enhancing the fiber cell elongation stage by regulating the expression of tubulin, sucrose synthase, and expansin genes [28]. The gene encoding ethylene insensitive 3-like protein was differentially expressed between TM-1 and ZMS12 [11]. Recently, investigations on the application of the relative genome in the analysis of function and evolution of the *EIL/EIN3* gene family have been reported [18,29,30]. However, there is still a lack of specific evolutionary relationships of the *EIL/EIN3* gene family in cotton. Fiber mutants are powerful tools for investigating the molecular mechanism and physiological process of fiber cell developmental stages. Mutant cotton analysis has accelerated the identification and characterization of interest genes or molecular elements associated with various stages of cotton fiber development [31,32,33,34]. To fill this gap, the evolutionary relationships of *EIL/EIN3* genes from cotton species, including *Gossypium hirsutum*, *Gossypium arboreum*, and *Gossypium raimondii* were analyzed, according to their tree topology and sequence similarity. Moreover, the expression levels of upland cotton *EIL/EIN3* genes were studied on a variety of fiber-cell developmental stages including 0, 3, 5, 8, and 10 days post anthesis (DPA). The results of this study provide fundamental evidence into the role of *EIL/EIN3* genes in cotton fiber development and will support future functional examination of this vital gene family. In cotton, the *EIL/EIN3* transcription factor plays an important role in the ethylene signaling pathway, which is involved in regulating the posttranscriptional level, whereas transcriptional regulations of *EIL/EIN3* are not yet fully known.

## 2. Materials and Methods

### 2.1. Gene Identification Procedures

The identification of the *EIL/EIN3* transcription factor family in three cotton genome protein sequences (*G. hirsutum*, *G. arboreum* and *G. raimondii*) were downloaded from cottongen database resources (https://www.cottongen.org/). To identify *EIL/EIN3* transcription factors in the cotton genome a BLASTP search was conducted using *Arabidopsis*, *P. trichocarpa*, maize, sorghum, and rice. *EIL/EIN3* protein sequences were identified in the Phytozome database (https://phytozome.jgi.doe.gov/pz/portal.html). The Hidden Markov Model profile of the *EIL/EIN3* protein domain (PF04873) was downloaded from the Pfam database resources (http://pfam.xfam.org) and used as query sequences to identify *EIL/EIN3* genes with an E-value < 10^−10^. To confirm the presence of the *EIL/EIN3* protein domain in the three cotton genome protein sequences, all putative protein sequences were scanned by the InterPro online tool (http://www.ebi.ac.uk/interpro/) and SMART database (http://smart.embl-heidelberg.de/) to validate the *EIL/EIN3* domain. Additionally, only the protein sequences with the *EIN3/EIL* domain were kept for further investigation. The physiochemical features (isoelectric point and molecular mass) of all the identified *EIL/EIN3* protein domains were examined by the ExPASy Server tool. Moreover, the prediction of subcellular location was done by WoLF PSORT (https://wolfpsort.hgc.jp/) and confirmed by Protein Prowler Subcellular Localization Predictor version 1.2 (http://bioinf.scmb.uq.edu.au:8080/pprowler_webapp_1-2/) and TargetP1.1 Server11 (http://www.cbs.dtu.dk/services/TargetP/). The physical locations of *EIN3/EIL* genes were determined through blastN queries against the three cotton genomes. The MapChart tool (https://mapchart.net/) was used to locate *EIN3/EIL* genes on the specific chromosome.

### 2.2. Phylogenetic Analysis of the EIL/EIN3 Protein Family

The ClustalW method was used to align full amino acid sequences of *EIL/EIN3* genes from *G. hirsutum*, *G. arboreum*, *G. raimondii*, *Arabidopsis thaliana*, *P. trichocarpa*, maize, sorghum, and rice. Moreover, a neighbor-joining (NJ) phylogeny method was conducted using MEGA 6.0 software (https://www.megasoftware.net/) with the following parameters: A Poisson model and 1000 bootstraps. The *EIL/EIN3* gene organizations (exon/intron) were created by Gene Structure Displayer Server 2.0 (http://gsds.cbi.pku.edu.cn/) according to their genomic DNA sequences [31]. The 10 conserved motifs of *EIN3/EIL* proteins were examined by the MEME program (http://meme-suite.org/).

### 2.3. Analysis of Cis-Acting Regulatory Elements and miRNA Targets

The *cis*-acting regulatory elements in each promoter sequence of *EIN3/EIL* genes (2 kb upstream of the translation starting site) were investigated using the PlantCARE database (http://bioinformatics.psb.ugent.be/webtools/plantcare/html/). The plant small RNA target analysis server (http://plantgrn.noble.org/psRNATarget/) [35] was used to predict the miRNAs target *G. hirsutum EIL/EIN3* genes.

### 2.4. Gene Ontology (GO) and Kyoto Encyclopedia of Genes and Genomes (KEGG) Pathway Analysis of EIL/EIN3 Genes

Gene Ontology (GO) was used to get information about the *EIL/EIN3* genes involvement in three ways: biological processes, cellular components, and molecular function. GO annotation analysis of cotton *EIN3/EIL* genes was done based on an OmicsBox/Blast2Go (https://www.biobam.com/omicsbox/). The KEGG (Kyoto Encyclopedia of Genes and Genomes) pathway was also used for a bioinformatics investigation that sought to examine the organismal and cellular functions of the *EIL/EIN3* genes. To visualize the KEGG-pathway-enrichment of cotton *EIL/EIN3* genes, *Arabidopsis* gene homologs for the identified *EIL/EIN3* proteins in cotton were utilized as input data for the David database (http://david.abcc.ncifcrf.gov/).

### 2.5. Protein Interaction Networks of EIL/EIN3 Genes

The investigation of protein–protein interactions simplifies the understanding of plant gene functions [36]. Consequently, to gain more insight into the role of *EIN3/EIL* genes in biological involvement through protein–protein interaction, the STRING online database (https://string-db.org/) was used to conduct the protein interaction networks.

### 2.6. Plant Material, RNA-seq, and Data Analysis

Upland cotton, Ligonlintless-1 mutant, and wild-type genotypes were planted in the experimental field at the ICR, Institute of Cotton Research, CAAS (Chinese Academy of Agricultural Sciences) under normal conditions. For self-pollination, cotton flowers before the anthesis day were tied and labeled. All cotton samples were collected with biological triplicates for each sample from both the Ligonlintless-1 mutant and wild-type at the following stages of cotton fiber development: 0, 3, 5, 8, and 10 DPA and cotton leaf. The collected cotton samples were quickly frozen in liquid nitrogen and kept at −80 °C until further analysis. RNA-seq data for upland cotton, Ligonlintless-1, and wild-type from cotton fiber developmental stages at 0, 3, and 8 DPA, and cotton leaf were used to identify the spatio-temporal expression levels of *G. hirsutum EIN3/EIL* genes. The raw reads were mapped to the upland reference genome [37]. The transcriptome assembly and expression counts were gained using STRINGTIE [38]. The FPKM value (fragments per kilo-base of exon per million fragments mapped) was used to calculate the gene expression pattern. The RT-qPCR analysis was also used to calculate the expression pattern of *G. hirsutum EIL/EIN3* genes associated with cotton fiber developmental stages, 0, 3, 5, 8, and 10 DPA. The SYBER premix ExTaq kit (TaKaRa, Osaka, Japan) and Applied Biosystems 7500 Real-Time PCR system (Applied Biosystems, Foster City, CA, USA) were used to conduct RT-qPCR experiments. The *G. hirsutum* constitutive β-actin gene was applied as a reference gene and special *EIL/EIN3* gene primers were employed for RT-qPCR. The following thermal cycle circumstances were applied: 95 °C for 2 min, 40 cycles of 95 °C for 5 s, and 60 °C for 34 s. The expression profiles for each gene were measured as the mean signal intensity across the three biological replicates. The Ct was used for the relative calculation of the input target number.

## 3. Results and Discussion

### 3.1. Gene Identification Procedures

To identify the *EIL/EIN3* transcription factor family in *G. hirsutum*, *G. raimondii*, and *G. arboreum*, two methods of the blast (local blast and HMM research) were used to search against the tree cotton genomes for E-values ≤ 10^−10^. Multiple sequence alignment was implemented to remove the redundant sequences of *EIL/EIN3* genes. Pfam (http://pfam.xfam.org/) and SMART (http://smart.embl-heidelberg.de/) databases were used to confirm the presence of the *EIL/EIN3* domain in cotton protein sequences. Finally, the redundant protein sequences were removed, resulting in 37 genes, including, 18, 10, and 9 *EIL/EIN3* genes in *G. hirsutum*, *G. raimondii*, and *G. arboreum*, respectively. The results of this study revealed that the EIN3/EIL transcription factor family is significantly smaller than other transcription factor families [39]. The same conclusion was reached in earlier reports in different plants [29,30,39]. Based on bioinformatics analyses of the studied species, an *EIN3* protein domain in each protein sequence of the *EIL/EIN3* gene members was found and localized in the nucleus, and various *EIL/EIN3* genes were found in higher plant genomes [15,29,39]. This indicated that the *EIL/EIN3* gene family may be involved in regulating various functions in different species. Additionally, the predicted nuclear positions may prove that the *EIL/EIN3* genes act as transcription factors. To obtain more insight into the possible role of the proteins encoded by the *EIL/EIN3* gene under examination, it is necessary to understand their physiochemical properties, for instance, the proteins can be divided based on their molecular weight and isoelectric point properties [40]. Through the enzyme association on the carrier, the bumper must have a pH value supporting electrostatic interactions with the surface of the carrier [40]. Additionally, in calculating any given protein family in plants, diverse physiological properties are investigated, for instance, in the sucrose synthase protein family in cotton, molecular weights and isoelectric points, among others were factored into the investigations [41]. The variations of an amino acid sequence of the cotton *EIL/EIN3* genes significantly varied in size and were divergent in physicochemical characteristics (Table 1). For example, the cotton *EIL/EIN3* proteins varied in length from 109 to 690 amino acids. Cotton *EIL/EIN3* proteins had a huge divergence in the isoelectric point (*pI*), ranging from 4.78 to 9.35. The molecular weight of the cotton *EIN3/EIL* proteins ranged from 13052.13 kDa to 77907.98 kDa (Table 1). WoLF PSORT, Protein Prowler Subcellular Localization Predictor, and TargetP1.1 Server analyses found that all of these cotton *EIL/EIN3* proteins, including soybean, *P. trichocarpa*, and *Arabidopsis EIL/EIN3* family proteins, were found to be localized in the nucleus compartment, which is in-line with their role in transcriptional regulation [39,42]. These results suggest that the *EIL/EIN3* protein family may be involved in the regulation of various aspects of plant growth and development.

To analyze the distribution of the *EIL/EIN3* genes on chromosomes on *G. hisurtum*, *G. raimondii*, and *G. arboreum*, a chromosome map was constructed based on three cotton genome sequences. In *G. hirsutum,* 17 genes were distributed across 13 of the 26 chromosomes while 1 gene was mapped to an unknown chromosome (scaffold). Two *EIL/EIN3* genes were mapped to chromosomes A_h_05, A_h_13, D_h_03, and D_h_13, and one *EIL/EIN3* gene was localized on chromosomes A_h_02, A_h_03, A_h_06, A_h_08, D_h_05, D_h_06, D_h_07, D_h_08, and D_h_12 (Table 1 and Appendix A). The distribution of *EIL/EIN3* genes within the At and Dt sub-genomes were consistent, with 8 and 9 *EIL/EIN3* genes located in the sub-genomes, respectively. Nevertheless, there is a strong indication of gene loss between the two loci sets of homologous chromosomes. For instance, chrA_h_03 has a single *EIN3/EIL* gene, but its homolog, chromosome D_h_03 has 2 *EIL/EIN3* genes. Likewise, chromosome A_h_05 has 2 *EI EIL/EIN3 N3/EIL* genes, whereas chromosome D_h_05 only has one gene. Almost similar numbers of putative EIN3/EIL were located on the At and Dt sub-genomes with 8 and 9 genes, respectively. In *G. raimondii,* two genes were mapped to chromosomes Chr D_5_03, Chr D_5_09, and Chr D_5_13 while one gene was localized on chromosomes Chr D_5_01, Chr D_5_04, Chr D_5_07, and Chr D_5_10. In *G. arboreum*, two genes were distributed on chromosomes ChrA_2_05 and Chr A_2_13, and one gene was mapped to chromosomes Chr A_2_01, Chr A_2_02, Chr A_2_06, and Chr A_2_08, with the remaining one gene localized on a scaffold region. Genome chromosomal position analyses revealed that the supposed 37 *EIL/EIN3* genes were not consistently distributed throughout all cotton chromosomes (Table 1 and Appendix A).

### 3.2. Gene Structure Analysis and Conserved Motif Analysis of EIL/EIN3 Genes

The variation in gene arrangement is the key factor involved in the evolution of multigene families [43,44,45]. To elucidate the organizational diversities of cotton *EIL/EIN3* genes, the exon/intron arrangements of these cotton genes were investigated. The result of the gene structure analysis found that most cotton *EIL/EIN3* genes (24) did not have introns, while 12 genes contained an intron. Of the 12 *EIL/EIN3* genes with introns, ten genes had one intron, one contained two introns, and the other contained four introns (Figure 1A). These last two genes were from *G. arboreum*, and it was speculated that some members of the *EIL/EIN3* genes lost extra introns during the course of hybridization. However, in the majority of *G. raimondii* genes with upstream and downstream fragments, it was noticed that gene members of *G. raimondii* reformed during the process of hybridization.

In order to obtain further insight into the conserved motifs, 10 motifs in cotton *EIL/EIN3* proteins were conducted using the MEME program (Figure 1B and Appendix A). A set of 37 *EIL/EIN3* protein sequences resulted in three common motif sequences: 1, 3, and 5 motifs were associated with the EIN3 (PF04873) domain, which was found to be located in the first half of the amino acid sequences (starting from 33 to 208 residues, approximately). In previous identification of conserved motifs in higher plants, it was found that motifs 1, 3, 4, 5, and 6 were related to the domain of EIN3/EILs [12,14,29,30]. Earlier studies reported that the first part of the *EIL/EIN3* gene was found to contain highly homologous sequences, which were associated with the domain of *EIN3* [15]. This was similarly confirmed in several recent works [29,42]. Additionally, some changes in the motif numbers and sequences were also mentioned in the first half of the N-terminus, except in the first ~80 residues for activity [17]. Moreover, the existence of two long conserved motif residues related to the EIN3 domain also implicated the significantly conserved motif organizations of cotton *EIL/EIN3.*

### 3.3. Phylogeny of EIL/EIN3 Members

Phylogenetic tree analysis was generated by MEGA6 software with the NJ method for 1000 bootstraps using 74 *EIL/EIN3* protein sequences from 18 *G. hirsutum*, 9 *G. arboreum,* 10 *G. raimondii,* 6 *Arabidopsis*, 7 *P. trichocarpa*, 6 maize, 3 sorghum, 3 *T. cacao,* and 12 soybean *EIL/EIN3* proteins (Figure 2). Phylogeny was applied to gather functional relationships for the supposed 74 *EIL/EIN3* genes. Phylogenetic tree analysis of the 74 *EIL/EIN3* protein sequences was divided into three main groups, labeled as groups A, B, and C, which varied in number from 16 to 38 *EIL/EIN3* genes (Figure 2). Group A had the largest number with 38 *EIL/EIN3* genes, followed by group B with 20, and group C with 16. The results of this study are in agreement with previous reports in other plants on classifications of these *EIL/EIN3* gene groupings [30,39,42]. The *EIL/EIN3* protein families in dicots and monocots were grouped in different sub-groups in our phylogenetic tree compared to previous findings. The current findings revealed that nine plant species presented into the three main groups of *EIL/EIN3* gene members, which provided a strong sign that the variance of these plants arose after the extension of the *EIL/EIN3* transcription factor gene family. The distribution of *EIL/EIN3* genes was much greater in the three *Gossypium* genomes than in other plant species. Furthermore, a unique observation was made, in which several clades only contained members of *EIL/EIN3* genes derived from a particular plant species. Some *EIL/EIN3* gene homologs were clustered by plant species within a sub-group, which referred to that plant species.

### 3.4. Cis-Regulatory Element Analysis

The presence of various *cis*-regulatory elements in promoter regions of *EIL/EIN3* genes may indicate that the functions of these genes are diverse. In order to explore *cis*-regulatory elements in the promoter regions of cotton *EIL/EIN3* genes, a 1500 bp upstream region of the transcription start site in each gene was identified and then searched for in the PlantCARE database. Various types of *cis*-regulatory elements were found in the promoter regions of cotton *EIL/EIN3* genes (Appendix A). Particularly, regulatory sequences such as CAATs, TATA boxes, TGACG-motifs, MYB recognition sites, and STREs (rapid stress response elements) were detected in all upland cotton *EIL/EIN3* genes. CAAT and TATA boxes are the main *cis*-acting elements found in the promoter sequences of transcriptional eukaryotic genes. In addition, the CAAT-box forms a binding position for RNA transcription factors and is involved in modulating the expression of genes [46,47,48]. The CAAT-box element plays an important role in regulating the nopaline synthase promoter [49]. Recently, the TATA-box was reported to harbor a binding site for histones or transcription factors, which supported its role in the transcriptional process [50,51]. STREs are involved in the primary transcriptional stress response to abiotic and biotic stresses in vivo [52]. Furthermore, several identified *cis*-regulatory elements were associated with hormone responses such as the CGTCA- and TGACG-motifs (*cis*-acting regulatory elements involved in MeJA-responsiveness), GARE-motif (*cis*-acting regulatory element involved in the gibberellin-responsive element), TGA-element (*cis*-acting regulatory element involved in auxin responsiveness), TCA-element (*cis*-acting element involved in salicylic acid responsiveness), and ABRE (*cis*-acting regulatory element involved in abscisic acid responsiveness) (Appendix A). In cotton ovules, it was noticed that abscisic acid is involved in the regulation of the cotton fiber developmental stages, along with auxins, ethylene, and gibberellins [53]. In general, plant hormones play an important role in the cotton fiber developmental stages such as initiation, elongation, and secondary cell wall development [54]. The result of this study also showed other *cis*-regulatory elements with diverse functions, such as improving the ability of the plant’s responses to drought and light, among others. Analysis of the identified *cis*-regulatory elements revealed that *EIL/EIN3* genes are related to various functions during cotton plant growth and development. The majority of *EIL/EIN3* gene promoters had a combination of various hormones related to *cis*-elements. These findings support that *EIL/EIN3* genes play a vital role in regulating diverse hormone signaling pathways, and suggest that these *EIL/EIN3* genes are transcriptionally controlled by a multicomplex network of hormones. The outcome of this investigation indicated that some *cis*-regulatory elements associated with *EIL/EIN3* genes might be involved in the regulation of cotton fiber developmental stages.

### 3.5. GO and KEGG Pathway Analysis of EIL/EIN3 Genes

The GO annotations and KEGG pathway analyses were conducted using the OmicsBox/Blast2Go tool and KEGG (Kyoto Encyclopedia of Genes and Genomes) pathway, and they presented the possible functions of *EIL/EIN3* proteins. Cotton *EIL/EIN3* genes were classified into two groups by GO annotation analysis according to their categories: biological processes and cellular components. The GO annotation results assigned the *EIL/EIN3* genes into 14 clusters of biological processes; a great number of *EIL/EIN3* genes were involved in the ethylene-activated signaling pathway, cellular macromolecule metabolic process, nitrogen compound metabolic process, primary metabolic process, and cellular response to iron ions (Figure 3A). In terms of the cellular component prediction of *EIL/EIN3* genes, they were significantly enriched in the nucleus, endoplasmic reticulum membrane, intracellular membrane-bounded organelles, and endoplasmic reticulum sub-compartment. The KEGG pathway analysis showed that these *EIL/EIN3* genes play roles in ethylene regulatory networks of plant hormone signal transduction (ath04075). Ethylene plays a key role in promoting cotton fiber elongation by increasing the expression of tubulin, sucrose synthase, and expansin genes [55]. The ethylene pathway is one of the most important biosynthetic pathways related to cotton fiber development during the elongation stage [56].

### 3.6. Protein–Protein Interaction Networks of EIL/EIN3 Genes

Protein interaction networks are significantly involved in regulation of various cellular functions, including metabolic pathways, signal transduction, and cell cycle development [57]. In animals, protein interaction networks regulate signal transduction, which affects developmental patterns, global homeostasis, normal physiology, and disease [58]. In plants, protein interaction networks associate with molecular mechanisms of biological processes such as signal transduction, cell cycle regulation, pattern formation, organ formation, and plant defense [59,60]. To understand the roles of the *EIL/EIN3* genes in regulating cotton fiber developmental processes, the STRING database was used to generate the protein–protein interaction networks (Figure 3B and Appendix A). The result of protein–protein interactions of the *EIL/EIN3* genes showed that the *EIN3* protein domains physically interact with the serine/threonine-protein kinase CTR1, signal transduction histidine kinase/ethylene sensor (ETR), and E3 ubiquitin ligase complexes (EBF1 and EBF2). Serine/threonine protein kinase plays an important role in a multiplicity of developmental pathways including cell migration, cell proliferation, and cytoskeleton regulation [61,62]. Activation by diverse effectors containing growth factor receptors results in a conformational change and consequent autophosphorylation on several serine/threonine residues [63]. CTR1 associates with the endoplasmic reticulum membranes in *Arabidopsis* as a result of its connections with ethylene receptors [64]. The function E3 ubiquitin ligase is more complicated in several cellular processes because of the E3 action based on the existence of different subunits [65]. Many E3 ubiquitin ligase complexes are key players in the regulation of physiological and developmental processes in plant biology [66,67,68]. The ubiquitin-proteasomal system regulates the degradation of various proteins in plant cells and disturbs a wide-range of cellular processes such as cell division, signal transduction, and immune responses [69]. The results showed that a member of the *EIL/EIN3* protein family might form functional transcriptional factor complexes, mediating the expression of ethylene-insensitive3-like/ethylene-insensitive3 (*EIL/EIN3*) protein genes in cotton fiber developmental stages.

### 3.7. Expression Level Analysis of Cotton EIL/EIN3 Genes

To further understand the expression pattern of *EIL/EIN3* genes in cotton fiber developmental stages and their potential functions in the early termination of the fiber elongation process, RNA-seq data, including three stages of fiber cell development (0, 3, and 8 DPA) and cotton leaf, from Ligonlintless-1 and its wild-type were generated by Illumina sequencing. It was shown that all *EIL/EIN3* genes were expressed at three stages (0, 3, and 8 DPA) of cotton fiber development and leaf using the FPKM value (Figure 4A). In addition, most *EIL/EIN3* genes showed a variation in expression levels in different cotton tissues. The results showed that the *EIL/EIN3* genes were differentially expressed in the initiation and elongation stages of fiber development and cotton leaf. To investigate whether the expression pattern of the EIL/EIN3 genes had a function in fiber developmental stages, we estimated the expression patterns of *EIL/EIN3* genes from each transcript between Ligonlintless-1 and wild-type. A total of 4, 11, and 10 genes were up-regulated, and 14, 7, and 8 genes were down-regulated in fiber developmental stages at 0, 3, and 8 DPA, respectively (Figure 4), which provides evidence of being directly or indirectly associated with a Ligonlintless-1 mutant and the wild-type during the cotton fiber developmental stages. In addition, at 3 and 8 DPA, more up-regulated *EIL/EIN3* genes than down-regulated *EIL/EIN3* genes in Ligonlintless-1 with comparison to wild-type implies that they could be playing a key role in regulating fiber development in mutation. Overall, the results of the RNA-seq data are consistent with the qRT-PCR analysis (Figure 4B). Data from this study show that some EIL/EIN3 transcription factors involved in regulating fiber cell initiation and elongation stages at 0, 3, 5, 8, and 10 DPA may be the possible reason for the early termination of fiber development in the mutant cotton fiber. This transcriptome analysis revealed differential expression levels of some *EIL/EIN3* genes at various developmental stages of fibers, an indication that *EIL/EIN3* genes may play an important role in regulating the metabolic pathways of cotton fiber development. This result is consistent with earlier reports showing that several genes encoding for ethylene production of cotton fibers were up-regulated in the Ligonlintless-1 mutant compared to wild-type at 5 and 7 DPA [70]. In this research, we found higher levels of ethylene produced during fiber development at 8 DPA in the Ligonlintless-1 mutant, possibly leading to early termination of cotton fiber development. Ethylene metabolism-associated genes are differentially expressed in cotton species during fiber developmental stages [71]. The expression patterns of EIL/EIN3 genes are an indicator of their involvement in various physiological and biochemical functions, which are directly associated with fiber development during initiation and elongation stages. The accumulation of ethylene in *G. raimondii* results in the reduction of cotton fiber cells [72]. The findings suggest that ethylene may play a critical role in fiber cell development during the initiation and elongation stages.

### 3.8. Prediction of miRNA Targets in Cotton EIL/EIN3 Genes during Cotton Fiber Development

Fiber developmental stages rely on the multicomplex regulation networks of numerous genes. The miRNAs play an important role in the regulation of cotton fiber development. The miRNAs are defined as part of the noncoding RNAs, which broadly play an important role in regulating various aspects of plant growth and development [73]. In cotton, miRNAs are involved in different features of growth such as fiber developmental stages and response to various abiotic stresses [74,75,76]. Previously, it was mentioned that several miRNAs were extremely expressed in the elongation stage of cotton fiber in Ligonlitless-1 (Li1) and Ligonlintless-2 (Li2) mutants, and their wild-type (WT) [77]. In order to determine the additional roles of *EIL/EIN3* genes, we submitted whole genes to the psRNATarget [78] for investigating if some *EIL/EIN3* genes were targeted by miRNAs. A total of 9 miRNAs (ghr-miR156a, ghr-miR156d, ghr-miR159a, ghr-miR394b, ghr-miR7491, ghr-miR7495a, ghr-miR7497, ghr-miR7502, and ghr-miR7504b) were differentially expressed in cotton fiber development during initiation and elongation stages (Table 2). These nine miRNAs targeted 11 *EIL/EIN3* genes (*Gh_A05G0871*, *Gh_A06G0989*, *Gh_A08G1750*, *Gh_A13G1864*, Gh_A13G2005, *Gh_D05G3883*, Gh_D06G1178, *Gh_D08G2099*, *Gh_D13G2252*, *Gh_D13G2404*, and *Gh_Sca006457G01*) and the great number of them are linked to cotton fiber developmental stages [75]. Several *EIL/EIN3* genes were targeted by more than one microRNA; for example, *Gh_A05G0871*, *Gh_D05G3883*, and *Gh_D08G2099* genes were targeted by 4 microRNAs, and the *Gh_A08G1750* gene was targeted by 2 microRNAs (Appendix A). In cotton, a number of miRNAs were differentially expressed at 8 DPA of cotton development, such as ghr-miR156a, ghr-miR156c, ghr-mi-R159, and ghr-mi-miR2949 [77]. In earlier reports, it was found that miR156 is involved in reducing fiber length in cotton [79]. In this study, we found five genes were repressed by ghr-mi-miR156, ghr-mi-miR159, and ghr-mi-miR2949, this could explain the early termination of fiber elongation in the Ligonlintless-1 mutant. Additionally, ghr-miR156a could play a crucial role in cotton fiber initiation [80]. Previously, it noted that miR166, miR167, miR172, and miR2949 were highly expressed in ovules [76]. In cotton, ghr-miR7496a was down-regulated, but ghr-miR7497 was up-regulated during fiber development in Ligonlintless-1 [33]. These results shows that some *EIL/EIN3* genes may directly play a crucial role in regulating fiber developmental stages (initiation and elongation stages). The interactions between EIL/EIN3 genes and microRNAs provide fundamental information on the regulation of gene expression.

## 4. Conclusions

The results of this work revealed insights into the *EIL/EIN3* transcription factor family in cotton, a representative cash crop. These results show that some *EIL/EIN3* genes were directly targeted by miRNAs related to cotton fiber developmental stages. Further examination with functional studies of *EIL/EIN3* transcription factors is needed to understand the regulation and interaction of various molecular pathways in enhancing cotton fiber development during the initiation and elongation stages. The findings of this study provided fundamental evidence into the role of *EIL/EIN3* genes, which will support future functional examination of various important biological molecular mechanisms of this vital gene family in cotton.

## Figures and Tables

**Figure 1 plants-09-00128-f001:**
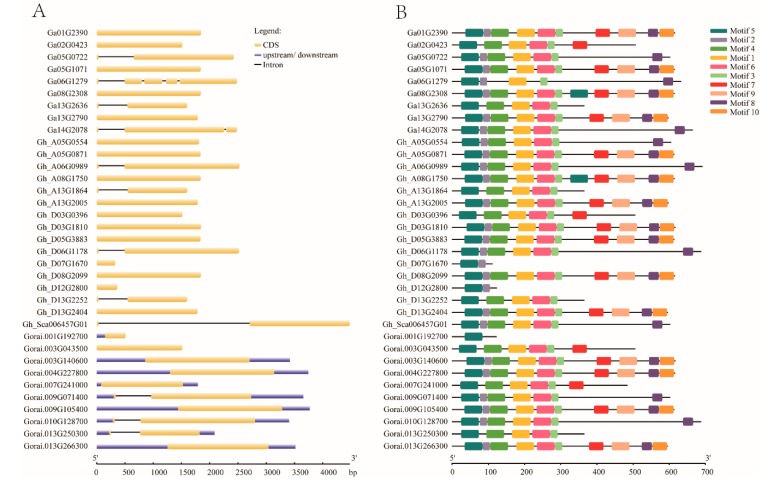
Structural features of *EIL/EIN3* transcription factor in cotton. (**A**) The exon–intron arrangement of *EIL/EIN3* genes. The blue shapes represent upstream/downstream, yellow shapes indicate CDS (exons), and black lines represent introns. (**B**) Ten motifs identified by the MEME tool are represented by colored boxes, and their consensus sequences are shown in Appendix A.

**Figure 2 plants-09-00128-f002:**
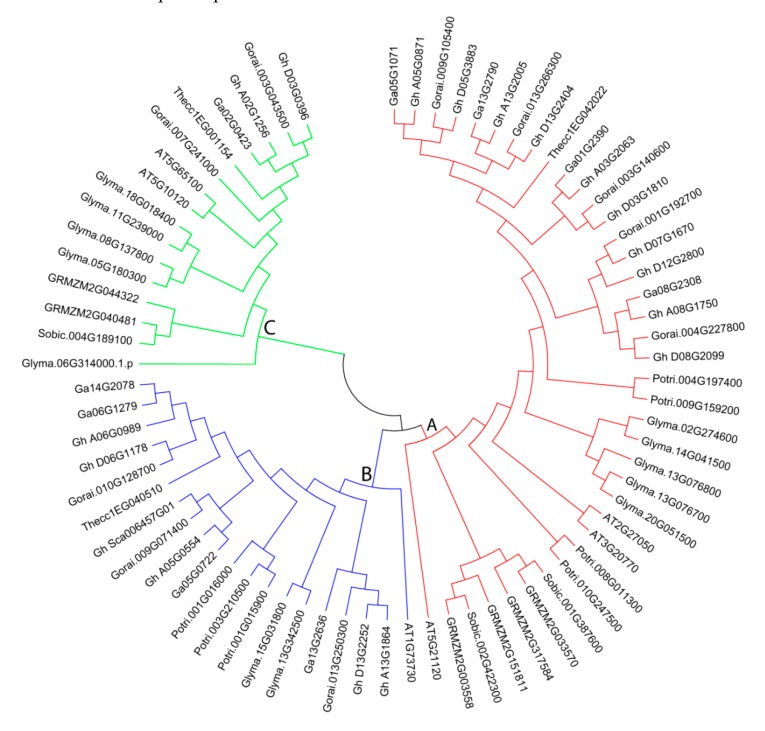
Phylogenetic tree relationships between 18 *G. hirsutum*, 9 *G. arboreum*, 10 *G. raimondii*, 6 *Arabidopsis*, 7 *P. trichocarpa*, 6 maize, 3 sorghum, 3 *T. cacao*, and 12 soybean EIL*/EIN3* proteins. The phylogenetic tree was created by the MEGA 6.0 program using the NJ (neighbor-joining) method. The bootstrap test was done with 1000 iterations. The three groups (A, B, and C) are shown in colors.

**Figure 3 plants-09-00128-f003:**
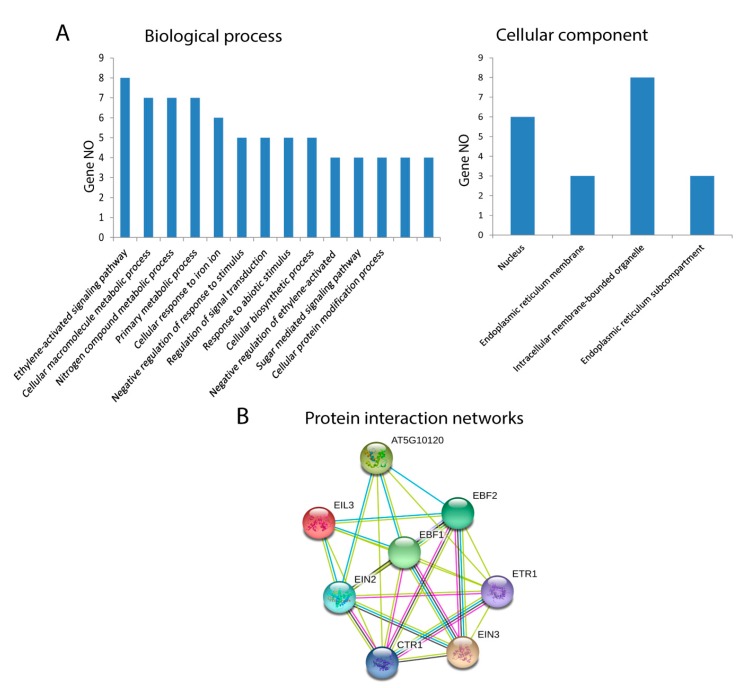
Functional analysis of *EIL/EIN3* proteins in upland cotton. (**A**) Gene ontology (GO) annotation results of the *EIL/EIN3* proteins. GO analysis of *EIL/EIN3* proteins tested for their function in biological processes, molecular functions, and cellular components. (**B**) Protein–protein interaction networks of *EIL/EIN3* transcription factors. Each node represents a protein and each edge represents an interaction between two proteins.

**Figure 4 plants-09-00128-f004:**
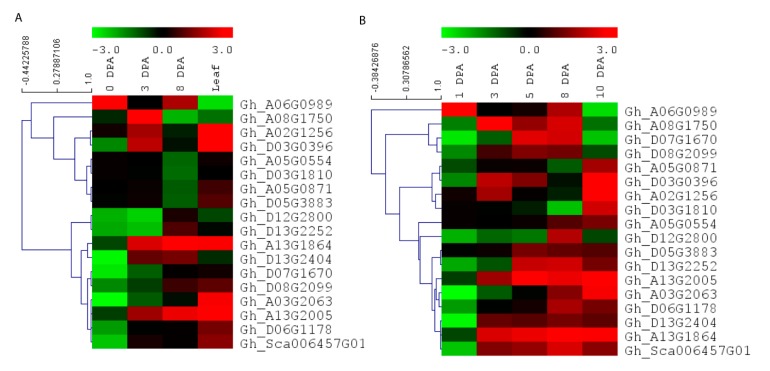
Expression patterns of *EIL/EIN3* genes in different stages of cotton fiber development and leaf. (**A**) Hierarchical clustering of expression profiles of the *EIL/EIN3* genes in cotton leaf and fiber development at 0, 3, and 8 days post anthesis (DPA) based on RNAseq data. (**B**) Hierarchical clustering of expression profiles of the *EIL/EIN3* genes in cotton fiber development at 0, 3, 5, 8, and 10 DPA using qRT-PCR analysis. The fold change values were log2 transformed. The standardized expression data were used to create heatmap with hierarchical clustering according to the Manhattan correlation with average linkage using the MeV software package. The color scale above the heatmap shows the expression levels: red indicates up-regulated while green indicates down-regulated genes.

**Table 1 plants-09-00128-t001:** Information about the *EIL/EIN3* gene family in cotton.

Gene ID	Protein Domain	Chro No.	Start	End	Protein Length	Isoelectric Point	Molecular Weight	Localization
Gh_A02G1256	EIN3	A_h_02	74,321,882	74,323,399	505	5.06	57,338.59	Nucleus
Gh_A03G2063	EIL3	A_h_03	76,519	78,363	603	5.47	68,277.79	Nucleus
Gh_A05G0554	EIL3	A_h_05	5,944,064	5,945,875	603	5.87	68,073.64	Nucleus
Gh_A05G0871	EIN3	A_h_05	8,658,831	8,660,669	612	5.38	69,300.14	Nucleus
Gh_A06G0989	EIL3	A_h_06	47,012,156	47,014,682	690	5.89	77,907.98	Nucleus
Gh_A08G1750	EIN3	A_h_08	97,487,518	97,489,359	613	5.53	69,551.64	Nucleus
Gh_A13G1864	EIL3	A_h_13	78,270,382	78,271,984	363	9.12	41,004.58	Nucleus
Gh_A13G2005	EIN3	A_h_13	79,505,190	79,506,980	596	5.2	67,093.59	Nucleus
Gh_D03G0396	EIL3	D_h_03	5,580,925	5,582,439	504	4.98	57,282.61	Nucleus
Gh_D03G1810	EIN3	D_h_03	45,097	46,944	615	5.6	69,643.55	Nucleus
Gh_D05G3883	EIN3	D_h_05	159,742	161,580	612	5.51	69,328.29	Nucleus
Gh_D06G1178	EIL3	D_h_06	29,244,232	29,246,749	686	5.97	77,587.57	Nucleus
Gh_D07G1670	EIN3	D_h_07	34,686,430	34,686,759	109	6.61	13,052.13	Nucleus
Gh_D08G2099	EIN3	D_h_08	59,885,901	59,887,745	614	5.58	69,713.77	Nucleus
Gh_D12G2800	EIN3	D_h_12	93,192	93,557	121	4.78	14,486.68	Nucleus
Gh_D13G2252	EIL3	D_h_13	58,573,978	58,575,579	363	9.11	40,835.48	Nucleus
Gh_D13G2404	EIN3	D_h_13	60,092,959	60,094,743	594	5.16	66,776.31	Nucleus
Gh_Sca006457G01	EIL3	scaffold6457	4225	8701	601	5.58	67,828.14	Nucleus
Gorai.001G192700	EIN3	Chr_5_01	32,645,047	32,645,559	120	5.03	15,060.47	Nucleus
Gorai.003G043500	EIL3	Chr_5_03	5,592,648	5,594,162	504	5.02	57,401.84	Nucleus
Gorai.003G140600	EIN3	Chr_5_03	40,374,524	40,377,941	615	5.54	70,282.28	Nucleus
Gorai.004G227800	EIN3	Chr_5_04	56,214,497	56,218,239	614	5.66	70,437.6	Nucleus
Gorai.007G241000	EIL3	Chr_5_07	34,165,596	34,167,386	482	5.33	56,260.32	Nucleus
Gorai.009G071400	EIL3	Chr_5_09	5,075,017	5,078,672	601	5.74	68,489.99	Nucleus
Gorai.009G105400	EIN3	Chr_5_09	7,642,067	7,645,836	612	5.56	70,090.19	Nucleus
Gorai.010G128700	EIL3	Chr_5_10	27,860,507	27,863,910	686	6.02	78,137.1	Nucleus
Gorai.013G250300	EIL3	Chr_5_13	56,813,952	56,816,039	363	9.35	41,574.39	Nucleus
Gorai.013G266300	EIN3	Chr_5_13	57,906,440	57,909,956	594	5.17	67,508.2	Nucleus
Ga01G2390	EIN3	Chr_2_01	106,813,982	106,815,826	614	5.58	69,535.33	Nucleus
Ga02G0423	EIL3	Chr_2_02	6,411,513	6,413,030	505	5.06	57,317.58	Nucleus
Ga05G0722	EIL3	Chr_2_05	6,269,696	6,272,122	601	5.66	67,740.02	Nucleus
Ga05G1071	EIN3	Chr_2_05	9,262,923	9,264,764	613	5.44	69,477.4	Nucleus
Ga06G1279	EIL3	Chr_2_06	53,063,302	53,065,786	631	5.83	71,738.04	Nucleus
Ga08G2308	EIN3	Chr_2_08	122,216,099	122,217,940	613	5.47	69,513.56	Nucleus
Ga13G2636	EIL3	Chr_2_13	122,248,180	122,249,780	363	8.61	40,688.25	Nucleus
Ga13G2790	EIN3	Chr_2_13	123,477,839	123,479,629	596	5.2	67,105.58	Nucleus
Ga14G2078	EIL3	tig00017874	3742	6220	663	5.88	75,328.17	Nucleus

**Table 2 plants-09-00128-t002:** Expression levels of miRNAs targeting *EIL/EIN3* genes in Ligonlintless-1 and wild-type.

miRNAs	Li1-0DPA	WT-0DPA	log2.Fold	q value	Li1-8DPA	WT-8DPA	log2.Fold	q value
ghr-miR156a	296.01	153.324	0.94906	5.26 × 10^−18^	254.3824	350.725	−0.4633	0.00071
ghr-miR156c	49.3349	51.108	−0.0509	0.057793	13.38854	58.4542	−2.1263	3.65 × 10^−8^
ghr-miR169a	0.77709	0.2271	−0.1263	0.089546	26.77709	29.2271	−0.1263	0.08955
ghr-miR394b	17267.2	22589.7	−0.3876	1.09 × 10^−27^	13910.7	18150	−0.3838	1.49 × 10^−52^
ghr-miR7491	296.01	1226.59	−2.0509	9.25 × 10^−107^	200.8282	233.817	−0.2194	0.05341
ghr-miR7495a	641.354	153.324	2.0645	3.97 × 10^−89^	240.9938	555.315	−1.2043	4.48 × 10^−24^
ghr-miR7497	49.3349	51.108	−0.0509	0.057793	13.38854	0	4.7429	1.49 × 10^−5^
ghr-miR7502	172.672	204.432	−0.2436	0.099541	66.94272	146.135	−1.1263	2.12 × 10^−7^
ghr-miR7504b	493.349	766.62	−0.6359	1.28 × 10^−7^	388.2678	409.179	−0.0757	0.05005

Note: Li1: Ligonlintless-1, WT: Wild-type, DPA: Days post anthesis, log2.fold: fold-change between Ligonlitless-1 and wild-type.

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
