# Peer review of "Investigation of the EIL/EIN3 Transcription Factor Gene Family Members and Their Expression Levels in the Early Stage of Cotton Fiber Development"

_plants, 2020, doi:10.3390/plants9010128_

Round 1

Reviewer 1 Report

Authors made some improvements from the last version. I still have a few comments and hope it could help authors to improve the manuscript. 1> please give units for column title in table 1; 2> please try to make font to be consistent in both size and family in figure 1A and 1B; 3> In figure 4, if you want to use phylogeny to show cluster of genes, please use it for both sub panels; 4> please mention the unit clearly for expression values in table 2

Author Response

Authors made some improvements from the last version. I still have a few comments and hope it could help authors to improve the manuscript. 1> please give units for column title in table 1; 2> please try to make font to be consistent in both size and family in figure 1A and 1B; 3> In figure 4, if you want to use phylogeny to show cluster of genes, please use it for both sub panels; 4> please mention the unit clearly for expression values in table 2.

Answer: Thank you for taking the time to send us your comments on manuscript. Any comment we receive is invaluable in helping us improve our manuscript. We have done that.

Reviewer 2 Report

This reviewer appreciates authors' dedication to address reviewers' comments and suggestions. After third round of revising this manuscript, this reviewer is comfortable to recommend this work getting published in its current form.

Author Response

This reviewer appreciates authors' dedication to address reviewers' comments and suggestions. After third round of revising this manuscript, this reviewer is comfortable to recommend this work getting published in its current form.

Answer: Thank you again for your positive comments on our manuscript.

This manuscript is a resubmission of an earlier submission. The following is a list of the peer review reports and author responses from that submission.

Round 1

Reviewer 1 Report

Major issues:

Citations need to be rechecked, thoroughly (see reasons below). There still are quite a few grammatical and sentence structure errors, only tracked a few of those here:

Introduction:

Page 1 [line 42]: The authors misunderstood the reviewer’s comment. Please remove ‘suffice’, using only ‘foundation’ in the sentence.

Page 1 [line 44]: ‘were found to be associated with’.

Page 2 [line 45]: ‘are one of the major transcription factors’

Page 2 [line 76]: please use ‘days post anthesis’ at DPA’s first use because readers may not understand the jargon.

Page 2 [line 78]: since ‘support future functional examination’ is already mentioned, there’s no need to say, ‘in the future’.

Results and Discussion

Page 4 [line 155]: E-value should be 10e-10.

Page 4 [line 168]: ‘for instance’ can be a complete sentence. Punctuation error in its current form.

Page 6 [lines 207-210]: the sentence starting with ‘the result of gene structure was found in the majority of cotton EIL/EIN3 genes’ is not clear. Perhaps “The result of gene structure analysis found that most cotton EIL/EIN3 genes did not have introns, whereas for two genes, one contained an intron and the other carried four introns”.

Problems with citations:

For example:

Page 3 [line 140]: Citation #37 is ‘RAMJI, D.P.; FOKA, P. CCAAT/enhancer-binding proteins: structure, function and regulation’. How is this citation related to ‘upland reference genome’? In fact, citation #59 is TM-1 genome. I found #59 cited for the following sentence:

"In plant, protein interaction networks accociate [please correct spelling] with molecular mechanisms of biological processes such as signal transduction, cell cycle regulation, pattern formation, organ formation and plant defense [59,60]"

I understand #37 and #59 are switched in this instance.

Page 4 [line 39]: STRINGTIE reference should be Pertea et al. (2015). Page 3 [line 113]: psRNATarget reference should be Dai et al. (2011). Current reference #35 has nothing to do with psRNATarget.

Reviewer 2 Report

I don't see a rebuttal letter for this revision, so I am not able to track detailed responses for my previous concerns. In this current version, there are still many places have obvious flaws that can be easily found out. Please carefully inspect those before submission and this current version is still not proper for publication as it is under the basic standard of a research paper. 

Many places still need careful inspection before submission, such as table 1, protein not prorien.  figure 1, I can not find subpanel A. Legends in figure 2, "three groups are shown in color...", but what does each individual color mean?  Figure 3A, you missed y axis title and readers won't know what these numbers are.  Figure 4, you should use the same font family and size for both of sub panels.  Table 2, you need to describe the unit for numbers listed in the table.